# Horizontal Stratified Air–Foam–Water Flows: Preliminary Modelling Attempts with OLGA

**William Ferretto** [1], **Igor Matteo Carraretto** [1,*], **Andrea Tiozzo** [2], **Marco Montini** [2] **and Luigi Pietro Maria Colombo** [1,*]

1    Department of Energy, Politecnico di Milano, Via Lambruschini 4, 20156 Milan, Italy
2    ENI S.p.A., Natural Resources Development, Operations & Energy Efficiency, Via Emilia 1, 20097 San Donato Milanese, Italy
\*    Correspondence: igormatteo.carraretto@polimi.it (I.M.C.); luigi.colombo@polimi.it (L.P.M.C.)

**Abstract:** Water accumulation is a major problem in the flow assurance of gas pipelines. To limit liquid loading issues, deliquification by means of surfactant injection is a promising alternative to the consolidated mechanical methods. However, the macroscopic behavior of foam pipe flow in the presence of other phases has barely been explored. The goal of this work was to propose an approach to simulate air–water–foam flows in horizontal pipes using OLGA by Schlumberger, an industry standard tool for the transient simulation of multiphase flow. The simulation results were compared with experimental data for 60 mm and 30 mm ID (Inner Diameter) horizontal pipelines. Preliminary validation for two-phase air–water flow was carried out, which showed that correct flow pattern recognition is essential to accurately reproduce the experimental data. Then, stratified air–foam–water flows were investigated, assuming different models for the foam local velocity distribution. Foam rheology was considered through the Herschel–Bulkley model with the yield stress varying in time due to foam decay. The results showed good agreement for a uniform velocity profile and fresh foam properties in the case of the 60 mm ID pipeline, whereas for the 30 mm ID, which was characterized by significantly higher velocities, a linear velocity profile and 2000 s foam aging provided the best agreement. In both cases, the pressure gradient was overestimated, and the mean absolute prediction error ranged from about 5% to 30%.

**Keywords:** foam; multiphase flow; pressure gradient; transient multiphase simulator

## 1. Introduction

Natural gas, among the traditional energy sources, has gained importance through the years due to its low carbon impact compared to those of coal and oil. In 2020, natural gas consumption, similar to all of the other traditional sources, suffered the economic consequences of the pandemic and decreased by 2.3% ($82 \cdot 10^9$ m$^3$); nevertheless, the share of gas in primary energy continued to increase, reaching 24.7% [1].

Among the major pipeline problems, water accumulation is of significant importance, especially at field late life. The presence of liquid leads to corrosion issues, frictional pressure drop increases, and the formation of gas hydrates. To limit liquid loading issues, different techniques have been developed: deliquification of the wellbore by means of surfactant injection is a relatively new approach and a promising alternative to the consolidated mechanical methods such as pigging or compressors [2]. The link between the microstructure properties and the macroscopic behavior of foam flow in pipes has been little investigated, and the studies on the latter are mainly of an experimental nature.

One of the earliest works on foam transport is the one by Briceno and Joseph [3], who investigated the flow characteristics of aqueous foams in a 5/8″ ID, 1.2-m-long Plexiglas® channel. The foam was generated in a packed bed through the introduction of air into a water stream containing dissolved surfactants. The authors observed seven flow patterns,

including stratified and slug flow, and established correlations among these patterns and foam quality and superficial velocity. Additionally, they found that the flow patterns observed in aqueous foam transport were similar to those seen in gas–liquid flow.

Gajbhiye and Kam [4] conducted research on foam characteristics in 0.5″ OD (Outer Diameter), 12 ft. horizontal pipes under a wide range of conditions, including variations in the pipe material, surfactant type, and concentration. The authors identified two foam flow regimes based on foam quality: the high-quality regime was associated with slug flow, while the low-quality regime was characterized by segregated or plug flow.

A more recent study by Amani et al. [5] investigated the dynamic behavior of foam flows in relation to foam characteristics. The authors conducted a comprehensive experimental campaign in a 4.3 m long, 44 mm ID vertical acrylic pipe at different flow rates and surfactant concentrations. The results showed significant changes in the flow regime when compared to the air–water case, with slug, churn, and annular flow observed and characterized through a Power Spectral Density (PSD) analysis of associated pressure fluctuations.

The literature most relevant to this paper, on the other hand, focuses on the issue of pipeline deliquification with foams.

Preliminary results on the conditions for foam formation were presented by Dall'Acqua et al. [6]. In particular, it was observed that the establishment of a localized slug regime at the elbow of a riser provided the necessary conditions for the foam formation within the pipe, opening the possibility for system deliquification.

Concerning the flow in horizontal pipes, Colombo et al. [7] performed an experimental study on a 20 m long 60 mm diameter Plexiglas horizontal pipeline. At low gas velocity, foam plugs/slugs with a shallow layer of liquid at the bottom were observed, while at higher gas velocity, a three-layer liquid–foam–gas stratified wavy flow was established. The liquid loading was estimated from the pressure gradient with the Taitel and Dukler [8] two-fluid model, resulting in an increase of the void fraction of 174% [7]. On the other hand, the liquid loading reduction was paired with a huge increase in the frictional pressure gradient, especially in the plug/slug flow regime.

Two other experimental campaigns were performed at the same experimental facility but with different inner pipe diameters, 60 mm and 30 mm. After an experimental air–water campaign, the surfactant was added to water and the foam quality, pressure drop, and liquid loading were measured. Two flow regimes have been identified: an intermittent plug flow at the lowest superficial gas velocity and a stratified wavy flow for the rest of the conditions.

Volovetskyi et al. [9] developed and tested a method to remove liquid from wells and pipelines using surfactants, addressing the problems of foam generation and destruction in the gas-liquid flow before it enters the gas treatment unit.

Zhang et al. [10] proposed a pigging approach with SDS (Sodium Dodecyl Sulfate) surfactant for foam drainage in slightly upward inclined (3°) gathering pipelines with wet gas production. The experimental investigation showed that the surfactant addition enabled suppressing slug flow and was favorable for liquid discharge.

Yin et al. [11] studied the flow patterns in air-deionized water flows with and without SDBS (Sodium Dodecyl Benzene Sulfonate) addition. The pipeline had an inner diameter of 50 mm and a length of 34 m with an adjustable inclination ($\pm$ 20°). Fully developed flow in an 8.5 m, 10° upward inclined section was investigated in order to visualize flow patterns and measure the pressure drop. SDBS surfactant solutions of 100, 400, and 800 ppm were selected to provide different degrees of foaming. The range of superficial velocities was 0.005–0.100 m s$^{-1}$ for the liquid and 2–30 m s$^{-1}$ for the gas. Flow patterns were classified as intermittent and segregated, whereas sub-flow patterns were identified by means of characteristic pressure drop fluctuations. The results were reported in maps showing the comparison between the flow with and without surfactant.

Zhang et al. [12] investigated aqueous foam drainage technology in horizontal upward pipelines at different inclination angles (5°, 10°, and 20°) to clarify the effects of foam on slug generation at the elbow. They concluded that in the presence of foam, the mechanism of slug generation was significantly modified since the wave aggregation mechanism was suppressed.

Yin et al. [13] analyzed the mechanism of liquid removal with surfactant in hilly terrain pipes by means of liquid holdup measurement and visualization by the image-quick closing valve method and wire mesh sensor method. An uphill test section (16.0 m long and 50.0 mm ID) was considered in a range of superficial velocities between 0.001 and 2.0 m s$^{-1}$ for water and between 2.0 and 14.0 m s$^{-1}$ for deionized water or 250 ppm SDS surfactant solution. A considerable reduction of the liquid loading was achieved using the surfactant, and transition from intermittent to segregated flow was observed. In particular, pseudo slug suppression promoted liquid unloading with continuous liquid loading in the pipeline.

Though further experimental investigations need to be carried out in order to provide a deeper understanding of several aspects of foam generation and transport, the available data can be used to validate modeling approaches, which at present are lacking, due to the complexity of the flow patterns and the non-Newtonian behavior of the foam.

To bridge this gap, as an initial attempt, this work aims to assess the capability of industrial simulation software to reproduce available data for air–water–foam flow in horizontal pipes. The industrial context makes use of general-purpose codes to address the technical issues pertaining to the so-called "flow assurance". The dedicated software is continually specialized in order to expand the simulation capability as far as new applications and/or the range of operating conditions are identified. In particular, OLGA by Schlumberger represents the industry standard tool for the transient simulation of multiphase petroleum production. For this reason, OLGA prediction performance was assessed with particular regard to the pressure drop, which was significantly increased by the presence of the foam compared to the gas–liquid flow without surfactant addition [7]. However, foam characteristics and dynamics are not implemented in the standard software; hence, this paper also proposes a methodology to include the foam as a new phase and to take into account its non-Newtonian rheology. Accordingly, the work was accomplished in two steps. First, two-phase air–water flow simulations were run to set a reference case and to understand the performance of OLGA with Newtonian fluid flow conditions. Finally, air–water–foam flow was modelled and compared against the available experimental results.

## 2. Materials and Methods: OLGA Model Description

### 2.1. Simulation Model

An OLGA simulation is controlled by different types of objects to form a simulation network. The network objects used in this work are of two types: *flowpath*, the pipeline through which the fluid mix flows; *node*, a boundary condition or connection point for two or more flowpaths. Each flowpath consists of a sequence of pipes divided into sections (control volumes) corresponding to the spatial mesh discretization in the numerical model. The spatial mesh applies flow variables at section boundaries and volume variables as average values taken at the middle of the section. Each flowpath must start and end at a node.

### 2.2. Model Basics

OLGA is a numerical simulator based on a three-fluid model where separate continuity equations are applied for gas, oil (or condensate), and water. Three momentum equations and one mixture energy equation are implemented. To close the system of seven equations, fluid properties and boundary and initial conditions are required. The model evolved from the basic architecture reported in [14], whereas the approach of three industrial simulators is compared in [15]. Several undetermined quantities, such as friction coefficients, are treated depending on the flow regime: two flow regimes, namely, stratified and slug flow, have been here analyzed.

For stratified flow, OLGA provides two different models: OLGA and OLGA HD. The former covers the stratified smooth and stratified wavy flow regimes using average bulk velocities and friction model closures. Wall frictions are modelled by a single-phase analogy with each phase layer being considered independent of the others. The OLGA HD flow model computes friction factors and mass flux terms using a 2-D (Two-Dimensional) velocity

distribution and friction factor as 1-D (One-Dimensional) models. The model agrees with the log law (for turbulence) at the pipe walls and a generalized log law at interfaces.

Concerning slug flow, the standard OLGA flow regime treats slugs through the so-called unit cell model. In this approach, slug flow is treated in an average manner where a control volume consists of an infinitely long train of identical, fully developed slug units. The slug flow is described by a combination of the other three flow regimes (stratified, annular, and bubbly).

The user controls the time integration by specifying the simulation period and time-step parameters. The spatial integration is performed on a user-defined grid.

### 2.3. Fluids and PVT

The compositional tracking model has been used in this work to provide fluid properties. The model combines the multiphase capabilities in OLGA with the customized calculations of the fluid properties. The PVT (Pressure Volume Temperature) package Multiflash, developed by KBC Infochem (London, UK), is used for the thermodynamic equilibrium calculations of the fluids.

### 2.4. Drilling Fluid Model

The drilling fluid model is a package of OLGA describing and tracking the flow of different fluids used in drilling operations, such as drilling muds. A drilling mud is a fluid used to aid the drilling of boreholes by providing hydrostatic pressure to prevent formation fluids from entering the wellbore, by cooling the drill bit and by carrying out drill cuttings. Typically, it consists of a base phase that may be water, oil or gas, and solid particles.

Two computational approaches are available in describing and modelling the mixing effects of different fluids: gas dissolvable and non-gas dissolvable. The former handles the dissolution of gas into the fluid, and the latter considers a mud phase change only through boiling or condensing.

OLGA allows a distinction between gas, oil-based, and water-based mud; the oil and water mud model computes the phase diagram through the specific parameters of the Antoine equation; the gas mud model describes the drilling fluid as a single-phase fluid. Mud properties can be identified through the OLGA Fluid Definition Tool by picking the compositional model.

### 2.5. Assumptions

Throughout this work, some assumptions have been made to model the different multiphase flows in the OLGA interface and to approach the experimental conditions as much as possible; the following are the most common.

- The pressure at the final node is fixed at the atmospheric pressure. The pipeline is adiabatic, and the mass fluxes' initial temperature is 25 °C.
- The full test section length has been considered in the simulations, and to account for flow development, an entry length of 4 m has been introduced. It has been verified that for such an entry length, a fully developed flow is obtained corresponding to a constant liquid loading.
- Concerning discretization, $10^{-1}$ m and $10^{-5}$ s are the respective spatial and time steps considered in the simulation. The selection of the former is dictated by the consideration that the experimentally observed slug units are at least three times longer than the distance between two mesh points. It is then not advised to increase the length of the mesh elements. Conversely, smaller mesh elements ($10^{-2}$) lead to percentage variations in both the liquid loading and the pressure drop lower than the experimental uncertainty (5%). Further, the computational time is obviously increased. Concerning the time step, it may be noted that a frequency of 100 kHz is much higher than that of any macroscopic structure flowing in the pipe. Eventually, a total simulation time of 2 h (to be on the safe side) is considered to reach steady-state conditions.

- Both the OLGA and OLGA HD flow models were initially considered. However, since the results showed a difference of less than about 1%, the former was adopted.
- The foam, being a flowing independent third phase, is modelled as a drilling fluid; specifically, a non-gas dissolvable gas mud model is selected. The choice of modelling the foam as a drilling fluid is ascribed to the absence of an equation of state capable of describing foam behavior and dissolution.

## 3. Results

### 3.1. Reference Air–Water Cases

First, air–water flow conditions were studied. Concerning the OLGA modelling, air and water phases were injected into two separate nodes from which two flowpaths exited and subsequently met in a mixing node (namely, the internal node); from there, the test section flowpath departed. The mass flow rates of both phases were fixed at the values retrieved from the experimental conditions.

For both cases (60 mm ID and 30 mm ID), the experimental test section consisted of a 24 m long horizontal Plexiglas pipeline equipped with a series of pressure taps [7,16]. The pressure drop and liquid holdup measurements were performed as follows:

- 60 mm ID: 24 operating conditions, eight air superficial velocities ($J_G = 0.77 \div 2.31$ m/s), and four water superficial velocities ($J_L = 0.03 \div 0.06$ m/s). All conditions exhibited a stratified flow regime, in agreement with the indications given by Mandhane [17] and Kong and Kim [18].
- 30 mm ID: 20 operating conditions, five air superficial velocities ($J_G = 2.67 \div 8.17$ m/s), and four water superficial velocities ($J_L = 0.12 \div 0.24$ m/s). All conditions exhibited a slug flow regime, in agreement with the indications given by Mandhane [17] and Kong and Kim [18]. Moreover, it is worth noting that the experimental points fell in the region of High Aerated Slug (HAS) as reported in the recent work by Arabi et al. [19].

The results of the experimental campaign were compared with the ones obtained through the simulations and are shown in a parity plot (Figure 1a). Specifically, two parameters were considered and reported in Table 1: the Mean Absolute Relative Deviation (MARD) and the Mean Relative Deviation (MRD), expressed as:

$$MARD = \frac{1}{n}\sum_{i=1}^{n}\left|\left(-\frac{dp}{dz}\right)_{model,i} - \left(-\frac{dp}{dz}\right)_{exp,i}\right|\left(-\frac{dp}{dz}\right)_{exp,i}^{-1}, \tag{1}$$

$$MRD = \frac{1}{n}\sum_{i=1}^{n}\left[\left(-\frac{dp}{dz}\right)_{model,i} - \left(-\frac{dp}{dz}\right)_{exp,i}\right]\left(-\frac{dp}{dz}\right)_{exp,i}^{-1}. \tag{2}$$

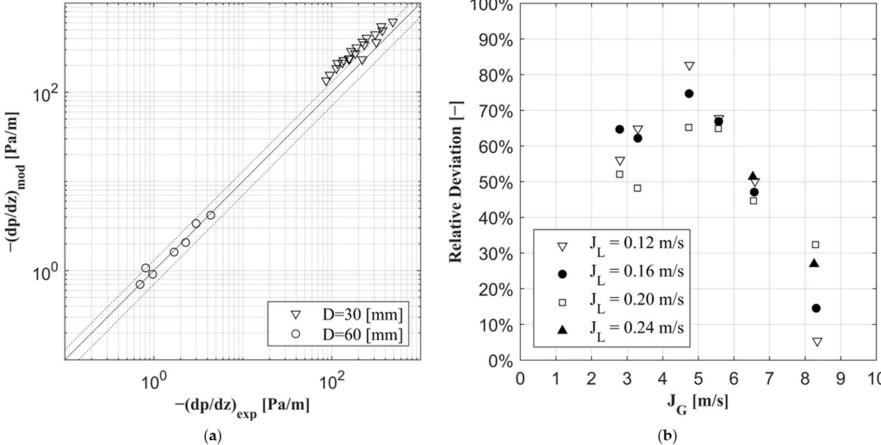

**Figure 1.** (**a**) OLGA pressure gradient log–log parity plot for 30 mm and 60 mm ID pipeline flows

(±30% error range is reported); (**b**) Relative deviation as a function of the gas superficial velocity at constant liquid superficial velocity for the 30 mm ID pipe (the points reported correspond to the flow pattern conditions correctly recognized by OLGA).

**Table 1.** MARD and MRD for the OLGA model for air–water flow.

| D [mm] | MARD [%] | MRD [%] |
|---|---|---|
| 60 | 10 | 3 |
| 30 | 52 | 52 |
| overall | 41 | 39 |

The prediction performance for the 60 mm ID pipe was quite good. However, as mentioned in Section 2.4, the closure equations consist of empirical models that are flow pattern dependent. On the other hand, the flow pattern is not an input parameter; rather, a flow pattern map is implemented in the algorithm. Hence, correct identification is essential to achieving a good predictive performance. Typically, flow conditions near a transition boundary may not be recognized correctly, leading to large prediction errors. Under most of the operating conditions for $J_L = 0.06$ m s$^{-1}$, slug flow was detected instead of stratified wavy, causing a significant overestimation of the pressure gradient.

For the 30 mm ID pipe, slug flow was identified in the whole range of operating conditions, but a systematic overestimation of the pressure gradient was observed as indicated by the fact that *MARD = MRD*. However, the performance prediction significantly improved as the gas superficial velocity increased, as seen in Figure 1b. As previously remarked, the larger error was associated with operating conditions lying closer to the transition between wavy and slug flows. Such an issue is still the object of intense research: superficial liquid velocity is recognized as influential in slug formation, but superficial gas velocity determines a variety of sub-regimes [20–22]; thus, the transition is not sharp. Hence, for both the 60 mm ID and 30 mm ID setups, under these operating conditions, the flow regime was challenging.

In summary, the performance prediction by OLGA was good provided that the flow regime was not transitional (i.e., fully established stratified or fully established slug flows).

*3.2. Foam Modelling*

Air–water–foam flows (AWFs) were simulated under the same conditions as those of the air–water flows.

### 3.2.1. Foam and Foam Flow Characteristic

A surfactant available for pipe deliquification purposes was adopted at the recommended concentration (0.3% in weight). The composition is provided in Table 2. More details, including a characterization of the static behavior, are reported in [7].

**Table 2.** Surfactant composition [7].

| Component | Concentration (wt. %) |
|---|---|
| Ammonium lauryl ether sulfate | 5–10 |
| Polyglycerol alkyl ethers | 50–60 |
| Propan-2-ol | 10–20 |
| 2-Butoxyethan-1-ol | <5 |

The foam was generated inside a mixing section where the liquid, previously enriched with surfactants, was vigorously mixed with air to start the foam formation process. Flow patterns in the considered superficial velocity range were framed into two main categories: plug flow and stratified flow. The former was characterized by a foamy head structure followed by a liquid layer surmounted by foam that obstructed the passage of gas (Figure 2a); as OLGA

could not identify this specific pattern, plug flow conditions were not considered. The latter occurred at higher gas superficial velocities and consisted of a liquid layer surmounted by foam and air layers (Figure 2b): this was the only flow regime analyzed.

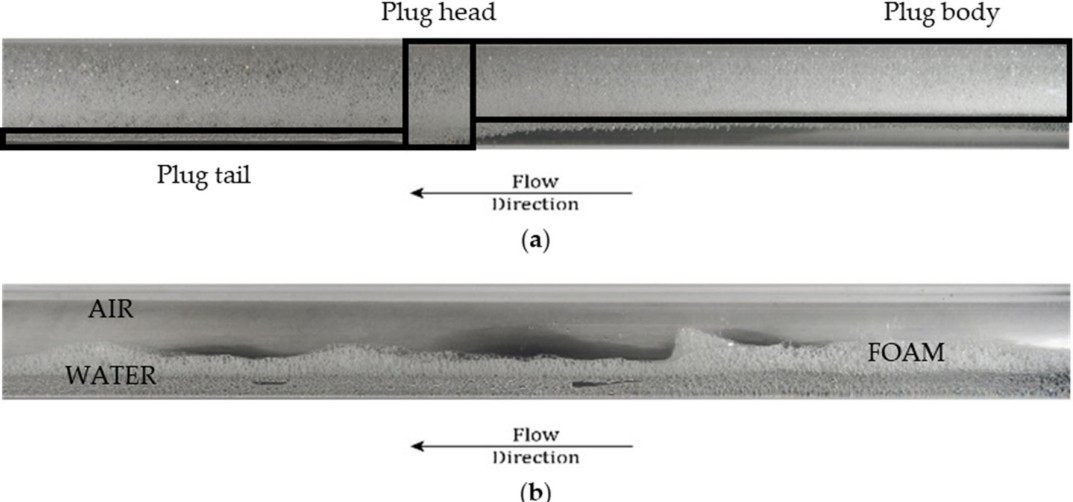

**Figure 2.** Examples of two foamy flow conditions (**a**) plug flow at $J_L$ = 0.03 m/s and $J_G$ = 0.41 m/s and (**b**) stratified flow at $J_L$ = 0.03 m/s and $J_G$ = 2.31 m/s (D = 60 mm).

The foam quality ($\phi$) was considered constant throughout all experimental conditions at a value of 0.97. This value was reported by the surfactant producer as the minimum value under which the foam would be unstable and corresponds to the average experimental value found in a previous campaign [7].

### 3.2.2. Foam Rheology

The Herschel–Bulkley model was considered for the foam. From previous experiments [23], foam rheology changes with time as the foam structure tends to collapse. Moreover, the dynamic conditions at high velocity enhance this process. This behavior influences the yield stress $\hat{\tau}$ while the consistency index $K$ and the flow index $n$ remain constant. The rheology parameters at different times from foam generation are reported in Table 3. Foams are subject to aging, which consists of coarsening due to both evaporation and drainage of the liquid laminae. The aging times in the table refer to static conditions, where the foam was generated in a container and was not displaced. If the foam moves in the presence of an airstream, it is expected that evaporation is promoted at least unless air is saturated with water vapor. Moreover, the action of the interfacial shear also promotes the breakage of the water laminae and the subsequent collapse of the foam. Hence, aging times shorter than those reported in Table 3 have to be expected for a moving foam.

**Table 3.** Herschel–Bulkley parameters for the foam at different aging times [23].

| t [s] | K [Pa·$^n$] | n [–] | $\hat{\tau}$ [Pa] |
|---|---|---|---|
| 0 | | | 8.50 |
| 500 | | | 6.50 |
| 1000 | 0.51 | 0.63 | 5.00 |
| 1500 | | | 3.77 |
| 2000 | | | 2.85 |

### 3.2.3. Mass Flow Rates

OLGA requires the direct specification of the mass flow rate for each phase: liquid, gas, and foam. Since from experimental data only the total gas and total liquid flow rates

are retrievable, some assumptions have to be made. Equations (3) and (4) report the overall mass balance for the liquid and gas phases.

$$J_L = \overline{U}_{LL}\varepsilon_{LL} + \overline{U}_{LF}\varepsilon_{LF}, \tag{3}$$

$$J_G = \overline{U}_A\varepsilon_A + \overline{U}_{GF}\varepsilon_{GF}. \tag{4}$$

The liquid loading $\varepsilon_{LL}$, the air fraction $\varepsilon_A$, and the foam fraction $\varepsilon_F$ were calculated from the flow picture analysis through ImageJ® software (v1.53n) [24]. The liquid fraction in the foam $\varepsilon_{LF}$ and the gas fraction in the foam $\varepsilon_{GF}$ were determined by assuming that the phases have the same velocity in the foam, i.e., $\overline{U}_{LF} = \overline{U}_{GF} = \overline{U}_F$.

The actual velocities $\overline{U}_{LL}$, $\overline{U}_A$ and $\overline{U}_F$ were retrieved by integrating the velocity profile over the corresponding layer area.

Air and water flows were assumed to be turbulent.

In particular, different shapes of the foam velocity profile are defined in OLGA, as follows:

- Uniform: the foam flows at a constant local velocity equal to the liquid local velocity at the water–foam interface.
- A power law with an exponent ranging from 1/7 (typical of turbulent flows) to $(n + 1)/n = 2.59$, i.e., the value that would arise from the laminar flow of a Herschel–Bulkley fluid (see Table 3), is used. Linear and parabolic velocity profiles were included in this range. Moreover, foam aging was considered as shown in Table 3.

Accordingly, the foam velocity profile and therefore $\overline{U}_{GF}$ and $\overline{U}_{LF}$ are linked to the air and liquid ones. Hence, inserting the unknowns, i.e., $\overline{U}_{LL}$, $\overline{U}_A$, $\overline{U}_{GF}$, and $\overline{U}_{LF}$, the velocity profile equations in (3) and (4) can be found through a trial-and-error procedure.

## 4. Discussion

Acceptable results were obtained only assuming a uniform and a linear velocity profile. In particular, the best prediction was achieved through the uniform velocity profile for the 60 mm ID pipe, through the linear one for the 30 mm ID pipe considering an aging period of t = 2000 s (Table 3).

Accordingly, the foam in the 60 mm ID pipe showed a solid-like behavior, being transported on top of the liquid layer with a negligible shearing action of the airstream above it. On the contrary, the superficial velocities in the 30 mm ID pipe were four times higher than those in the 60 mm ID pipe, which had two consequences: first, the shearing action on the foam layer was more pronounced and a velocity gradient developed in it; second, foam aging was expected to be significant. The prediction performance is summarized in Table 4.

**Table 4.** MARD and MRD for the OLGA model for air–water–foam flow.

| Model | MARD [%] | MRD [%] | Model | MARD [%] | MRD [%] |
|---|---|---|---|---|---|
| 60-UNI | 31 | 29 | 30-UNI | 39 | 30 |
| 60-LIN | 267 | 267 | 30-LIN-t = 0 s | 114 | 114 |
| - | - | - | 30-LIN-t = 1500 s | 26 | 26 |
| - | - | - | 30-LIN-t = 2000 s | 21 | 5 |

The simulated characteristics of the foam layer seemed consistent with the non-Newtonian behavior of the foam under certain simplifying assumptions. Namely, in a planar layer of local thickness $x$, moving along the direction $z$ (Figure 3), the shear stress according to the Herschel–Bulkley model is given as

$$\tau = \hat{\tau} + K\left(\frac{dU_F}{dx}\right)^n \tag{5}$$

where $\hat{\tau}$ is the yield stress. If $\tau < \hat{\tau}$, the fluid behaves like a solid body.

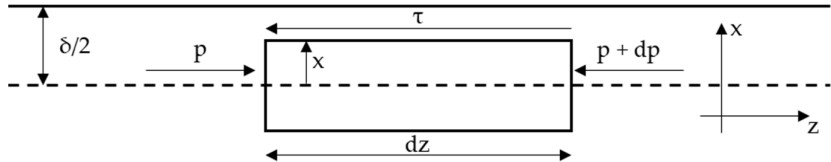

**Figure 3.** Control volume for the momentum balance in a planar layer.

The momentum balance for the layer is

$$\tau = -\frac{dp}{dz} \cdot x \tag{6}$$

The velocity gradient vanishes if $\tau = \hat{\tau}$. Accordingly, the thickness, $\delta_U$, of the layer in which uniform flow sets up is evaluated as

$$\hat{\tau} = -\frac{dp}{dz}\frac{\delta_U}{2} \text{ hence } \delta_U = \frac{2\hat{\tau}}{-dp/dz} \tag{7}$$

On the other hand, for $\tau \geq \hat{\tau}$, a velocity gradient develops in the layer.

Accordingly, if $\delta = \delta_U$, the foam layer is dragged similar to a solid object floating on the liquid, whereas if $\delta > \delta_U$, a velocity profile develops. Approximate estimates of $\delta_U$ were obtained taking the experimental values of both the yield stress and the pressure gradient, leading to a range between 10 mm and 25 mm for the 30 mm ID pipe and between 50 mm and 60 mm for the 60 mm ID pipe, which seems consistent with visual observations of the foam layer thickness (see a typical case in Figure 4). In particular, the 30 mm ID pipe most often showed $\delta > \delta_U$ whereas for the 60 mm ID pipe, $\delta \approx \delta_U$. This result is consistent with the outcome of the simulations.

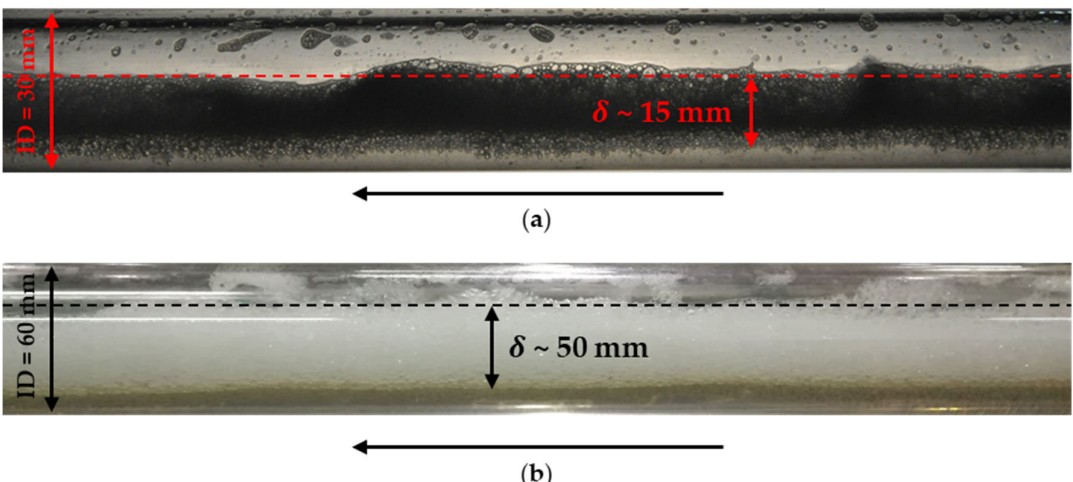

**Figure 4.** Qualitative comparison with visual observations: (**a**) ID = 30 mm and $J_L$ = 0.24 m/s, $J_G$ = 2.80 m/s, and $\delta_U$ = 12 mm, (**b**) ID = 60 mm and $J_L$ = 0.06 m/s, $J_G$ = 0.41 m/s, and $\delta_U$ = 53 mm.

Turning the attention to the effectiveness in the use of the foam to reduce the liquid loading, a deliquification parameter was introduced as the ratio of the void fraction with and without the foam at the same air and water superficial velocities:

$$\Delta = \frac{\varepsilon_{AWF}}{\varepsilon_{AW}} \tag{8}$$

The result is shown in Figure 5, where $\Delta$ is reported against the volume quality. It is evident that a significant reduction of the liquid loading took place, which was, however, more evident in the 60 mm ID pipe characterized by lower phase velocities. In both cases,

$\Delta$ decreased as the volume quality increased. This suggests that foam is more effective in reducing the liquid load at larger values of the water cut, which confirms the attractiveness of this deliquification technique. Accordingly, the percentage benefit ($\Delta$ 1) ranged between about 25% and 80% for the 60 mm ID pipe, whereas the 30 mm ID pipe showed lower values, in the range 15% to 20%.

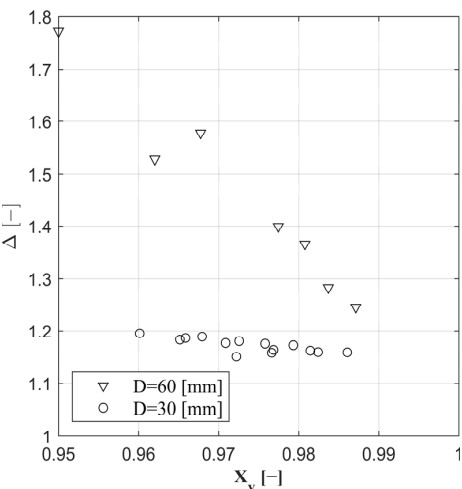

**Figure 5.** $\Delta$ vs. volume quality for the two pipelines tested.

Moreover, the simulated void fractions were successfully described in the frame of the drift–flux model [25,26]. Figure 6 shows the comparison between the flow without foam (Figure 6a) and with foam (Figure 6b), reporting the actual gas velocity against the mixture velocity. Regardless of the pipe diameter, the unique linear relationship is characterized by two parameters: the slope of the straight line, named the distribution parameter ($C_0$), which accounts for the non-uniformity of the flow in terms of both the local velocity and phase distribution; the intercept, named the drift velocity ($U_{GJ}$), which accounts for the difference between the gas and the mixture velocity. Such a relationship can be turned into a simple correlation between the void fraction and the volume quality [25]:

$$\varepsilon = \frac{x_v J}{C_0 J + U_{GJ}} \tag{9}$$

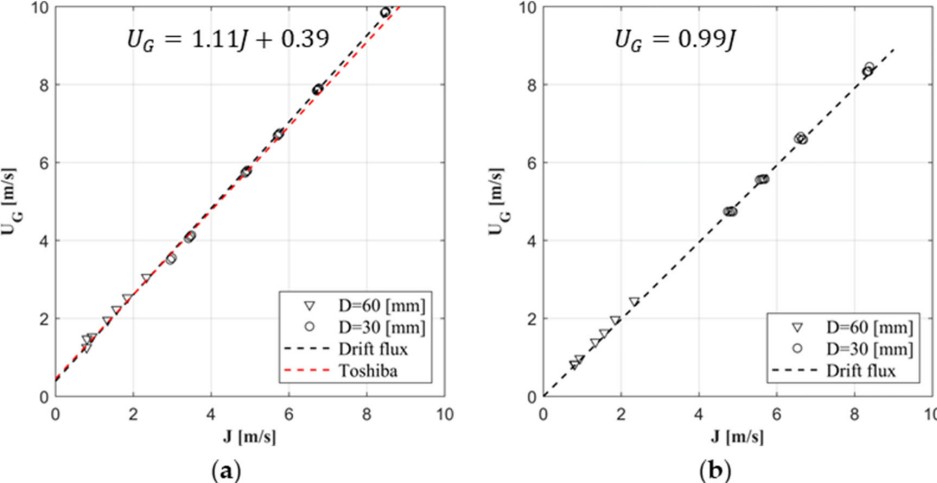

**Figure 6.** (**a**) AW flow results compared with the Drift–Flux and Toshiba models (Toshiba MRD = 2.63% and MARD = −1.21%) and (**b**) AWF flow results compared with Drift–Flux model.

The simulation of the stratified air–water flow without foam agreed quite well with the Toshiba drift–flux model (Equation (10)), which is reported in [26] as the best predictor over a broad experimental dataset regardless of the pipe inclination (Figure 6a).

$$\varepsilon = \frac{x_v J}{1.08 J + 0.45} \tag{10}$$

On the other hand, the effect of the foam layer formation was seen to homogenize the flow ($C_0 \cong 1$, $U_{GJ} = 0$, see Figure 6b). This is consistent with the experimental findings reported in [27], where the same trend was observed. Moreover, the considerations reported above about the foam dynamics also suggest that the foam layer exhibits an almost flat velocity profile under most of the simulated conditions, which strengthens the tendency to homogeneous flow.

## 5. Conclusions and Perspectives

The studies on foam flow behavior are mainly experimental, and no established modelling procedure has been developed so far. This work has shown a possible path in foam modelling using the software OLGA by Schlumberger for stratified flow patterns compatible with a significant range of operating conditions in the framework of pipeline deliquification.

Foam was treated as a non-Newtonian Herschel–Bulkley fluid floating on the liquid phase and sheared by the airstream on top.

The comparison of the simulated pressure drop with the available experimental data showed that foam aging is a crucial factor. In the lowest range of the mixture superficial velocity, corresponding to the 60 mm ID pipe, aging was negligible and the Herschel–Bulkley parameters for a "fresh" foam were suitable. On the other hand, when the mixture velocity quadrupled, i.e., in the 30 mm ID pipe, significant foam aging took place, possibly owing to the breakage of foam cells due to the increased shear, and the Herschel–Bulkley parameters for static aging times longer than 1500 s of the foam had to be adopted to acceptably predict the pressure drop.

It is then understood that a complete rheological (elasto-viscoplastic) characterization of the foam is crucial to properly simulate the flow in OLGA.

In addition, particular attention was paid to select the most appropriate velocity profile in the foam layer. The analysis showed that for the 60 mm ID setup, a uniform velocity profile better described the foam solid-like behavior (MARD = 31% and MRD = 29%). On the other hand, for the 30 mm ID setup, a linear velocity profile better described the shearing action, and consideration of the foam aging enabled MARD = 21% and MRD = 5% to be obtained.

From the point of view of phase distributions, the flow in the presence of foam exhibited a significantly higher void fraction than that without foam at the same superficial velocities of air and water: the percentage increase ranged from 15% to 80%, and it increased as the volume quality decreased, which confirms the effectiveness of the foam in reducing the liquid loading. Moreover, the implementation of the drift–flux model showed a tendency towards flow homogeneity in agreement with the experimental findings.

Future developments should extend the range of simulated conditions to include foam plugs and should assess the limitations in the use of commercial software.

**Author Contributions:** Conceptualization, W.F., I.M.C., M.M. and L.P.M.C.; methodology, W.F., I.M.C. and L.P.M.C.; software, A.T. and M.M.; validation, M.M. and L.P.M.C.; formal analysis, W.F., I.M.C. and L.P.M.C.; resources, I.M.C. and L.P.M.C.; data curation, W.F.; writing—original draft preparation, W.F. and I.M.C.; writing—review and editing, A.T., M.M. and L.P.M.C.; visualization, W.F. and I.M.C.; supervision, M.M. and L.P.M.C. All authors have read and agreed to the published version of the manuscript.

**Funding:** This research received no external funding.

**Data Availability Statement:** Data are available upon request.

**Acknowledgments:** The authors acknowledge Chimec S.p.A. for having provided the surfactant used during the experimental campaigns.

**Conflicts of Interest:** The authors declare no conflict of interest.

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
