# Peer review of "Horizontal Stratified Air–Foam–Water Flows: Preliminary Modelling Attempts with OLGA"

_fluids, doi:10.3390/fluids8030089_

Round 1

Reviewer 1 Report

In this paper, the authors have studied the feasibility of utilization of OLGA software in order to simulate the behavior of stratified air-foam-water flows. The simulations performed were compared with experimental results. Knowing that OLGA is not made to simulate this kind of multiphase flow, this works can be considered as a first step to propose an approach for this purpose. The manuscript is interesting and it is well written. Nevertheless, some points should be clarified and some results should also be more discussed. Therefore, I recommend major revision.

Please find below the points that should be addressed:

1/ Line 13: '..was very little investigated', it is preferable to change this sentence.

2/ Line 13, ..to find..', I think that the verb to propose is more suitable for this case.

3/ Please add 'transient multiphase simulator' as keyword.

4/ Paragraph between lines 55 and 60 is not clear. Is refers to the Ref. [4]? I think that the Colombo et al. [4] have performed experiments using only 60 mm ID pipe. This paragraph should be better explained.

5/ Concerned the experimental investigations carried out on air-foam-water multiphase flow and which are summarized in the introduction, I think that the introduction can be really improved if the authors summarize the goal behind all these experimental investigations.

6/ Is it possible to mention the work of Amani et al. (2020)?

Amani, P., Hurter, S., Rudolph, V., & Firouzi, M. (2020). Comparison of flow dynamics of air-water flows with foam flows in vertical pipes. Experimental Thermal and Fluid Science119, 110216.

7/ Line 96, replace '..is to simulate..' by 'is to propose a methodology in order to simulate' for example. The goal behind my comment is to better highlight the originality of your work.

8/ Lines 158 and 159, I think that it is better if the authors explain that these assumptions were be taken in order to get closer to the experimental conditions.

9/ In line 163, Why did the authors consider 4 m as entry length?

10/ Lines 167-169, is it possible to show together both results obtained with OLGA and OLGA HD? It can be really appreciated if the authors add this graph.), 

11/ The experimental results have been obtained using experimental setup, why this experimental ring was not described? I think that an additional section should be added in order to explain the experimental methodology. The paper will be improved.

12/ Is it possible to represent the experimental conditions in available flow maps in the open literature? I strongly the flow map of Mandhane et al. (1974) and Kong and Kim (2017). In order to get closer to the pipe diameter used in the present experiments, the authors can use the more recent flow maps of Arabi et al. (2022) and Wang et al. (1997) developed for 30 mm and 50 mm ID pipe, respectively.

Arabi, A., Salhi, Y., Zenati, Y., Si-Ahmed, E. K., & Legrand, J. Analysis of Gas-Liquid Intermittent Flow Sub-Regimes by Pressure Drop Signal Fluctuations. Proceedings of the 7 th World Congress on Momentum, Heat and Mass Transfer (MHMT'22) Virtual Conference – April 10 – 12, 2022

Kong, R., & Kim, S. (2017). Characterization of horizontal air–water two-phase flow. Nuclear Engineering and Design312, 266-276.

Mandhane, J. M., Gregory, G. A., & Aziz, K. (1974). A flow pattern map for gas—liquid flow in horizontal pipes. International journal of multiphase flow1(4), 537-553.

Wang, Y., Liu, Z., Chang, Y., Zhao, X., & Guo, L. (2019). Experimental study of gas-liquid two-phase wavy stratified flow in horizontal pipe at high pressure. International Journal of Heat and Mass Transfer143, 118537.

13/ Lines 198-201, the authors can discuss briefly about the difficulty to predict the transition from stratified to slug flow in order to better explain the difficulty for the software to predict the flow patterns in this region. I recommend to read the papers of Thaker and Banerjee (2015), Dinaryanto et al. (2017) and Arabi et al. (2021) 

Arabi, A., Salhi, Y., Bouderbal, A., Zenati, Y., Si-Ahmed, E. K., & Legrand, J. (2021). Onset of intermittent flow: Visualization of flow structures. Oil & Gas Science and Technology–Revue d’IFP Energies nouvelles76, 27.

Dinaryanto, O., Prayitno, Y. A. K., Majid, A. I., Hudaya, A. Z., Nusirwan, Y. A., & Widyaparaga, A. (2017). Experimental investigation on the initiation and flow development of gas-liquid slug two-phase flow in a horizontal pipe. Experimental Thermal and Fluid Science81, 93-108.

Thaker, J., & Banerjee, J. (2015). Characterization of two-phase slug flow sub-regimes using flow visualization. Journal of Petroleum Science and Engineering135, 561-576.

14/ Lines 208-209, explain the sentence '...provided that the flow regime is not transitional'

15/ In Fig. 1, why did you choose ±30% as tolerable error range?

Cai, Q., D'Auria, F., Umminger, K., Bestion, D., & Shan, J. (2022). Prioritizing pressure drop research in nuclear thermal hydraulics. Progress in Nuclear Energy153, 104358.

16/ In Fig. 1, you have mentioned in the section 3.1 that 24 and 20 operating conditions were performed for 60 and 30 mm ID pipes. Meanwhile, Fig. 1.a contains only approximatively 27 points. Is it possible to represent all the experimental points? 

17/ Fig.1, in order to highlight more the discussion of Fig. 2.b, is it possible to represent the operating conditions of experiments carried out on 30 mm ID pipe on a flow map using Jl and Jg as coordinates by distinguishing the points when more and less 30% deviations were observed?

18/ Lines 241-242, did you choose the same foam that used in the Ref. [12]?

19/ Lines 260-261, explain more about the picture analysis techniques used. You can refer to some published papers.

20/ Line 330, the Ref. [13] is not present in the references list.

21/ Paragraph between lines 338 and 345, explain that a same fit was used for both pipe diameters. for drift flux modelling. 

22/ It is preferable to add the correlation of Toshiba with its reference.

23/ Section 5, I strongly recommend to add 'perspectives' or 'recommendations for future works' with the conclusions.

24/ Move the paragraph between lines 366 and 367 in the end of the section 5. Indeed, it is a recommendation for future studies. 

Author Response

Dear Reviewer,

Thank you for your valuable suggestions to improve our work.

Please find attached our detailed response.

Best regards,

The Authors

Reviewer 2 Report

This work was to find an approach to simulate air-water-foam flows in horizontal pipes with OLGA by Schlumberger, an industry-standard tool for transient multiphase flow simulation. This study is ranged with the fluids journal's aim and scope. The paper's topic is important and draws interest from the readers. However, several key issues must be addressed before the manuscript can be published. Below are some comments to help the authors to improve the paper's quality.

 1.     The introduction should focus on the literature review of horizontal stratified air-foam-water flows. Therefore,  I suggest the authors add some references and rewrite the introduction by identifying the novelty/gap between this study and current research trends. The paper must offer a distinct novelty to the reader to be considered for publication.

 2.     Please explain why you use the OLGA software compared to other numerical models. What is the strong point of using the  OLGA simulation model?

3.     The simulation model should be presented in detail, including the numerical method, a schematic figure of the geometry model, and boundary conditions. Moreover, it is better to present a sensitivity analysis for simulation.

4. The results and discussion should be presented more profoundly and scientifically and need to compare to previous studies. Moreover, the limitation of calculation results should be stated in the paper because only a specific model and a boundary condition are selected. Is the presented OLGA model only valid for specific flow regimes (refer to table 1) and surfactant composition (refer to table 2)? It is better to add the geometry effect discussion due to the different pipe diameters and other flow regimes used.

5.      The conclusion needs to be written more technically

Author Response

Dear Reviewer,

Thank you very much for your valuable suggestions to improve the quality of our paper.

Please find attached the answers to your concerns.

Best regards,

The Authors

Round 2

Reviewer 1 Report

The authors made the requested corrections. I recommend to accept the manuscript.

Reviewer 2 Report

The authors have accommodated the reviewer's suggestion. The paper can be accepted now.